# Baseline Behavioral Data and Behavioral Correlates of Disturbance for the Lake Oku Clawed Frog (*Xenopus longipes*)

**Jemma E. Dias** [1], **Charlotte Ellis** [2], **Tessa E. Smith** [1], **Charlotte A. Hosie** [1], **Benjamin Tapley** [2] and **Christopher J. Michaels** [2,*]

[1]   Department of Biological Sciences, University of Chester, Parkgate Road, Chester CH1 4BJ, UK; jemmaedias@gmail.com (J.E.D.); tessa.smith@chester.ac.uk (T.E.S.); l.hosie@chester.ac.uk (C.A.H.)

[2]   Zoological Society of London Outer Circle, Regent's Park, London NW1 4RY, UK; charlotte.ellis@zsl.org (C.E.); ben.tapley@zsl.org (B.T.)

\*   Correspondence: christopher.michaels@zsl.org; Tel.: +44-2074496430

**Abstract:** Animal behavior and welfare science can form the basis of zoo animal management. However, even basic behavioral data are lacking for the majority of amphibian species, and species-specific research is required to inform management. Our goal was to develop the first ethogram for the critically endangered frog *Xenopus longipes* through observation of a captive population of 24 frogs. The ethogram was applied to produce a diurnal activity budget and to measure the behavioral impact of a routine health check where frogs were restrained. In the activity budget, frogs spent the vast majority of time swimming, resting in small amounts of time devoted to feeding, foraging, breathing, and (in males) amplexus. Using linear mixed models, we found no effect of time of day or sex on baseline behavior, other than for breathing, which had a greater duration in females. Linear mixed models indicated significant effects of the health check on duration of swimming, resting, foraging, feeding, and breathing behaviors for all frogs. This indicates a welfare trade-off associated with veterinary monitoring and highlights the importance of non-invasive monitoring where possible, as well as providing candidates for behavioral monitoring of acute stress. This investigation has provided the first behavioral data for this species which can be applied to future research regarding husbandry and management practices.

**Keywords:** amphibian; behavior; welfare; zoo research

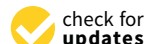



## 1. Introduction

Animal welfare is a central component of the management of animals in captivity, yet the basic tools to properly assess it are absent for many species [1]. A holistic understanding of behavior [2,3] alongside a scientific framework [4] can facilitate welfare management, but this requires species-specific data. Although welfare may be partly measured through the use of stress hormone analyses [5], behavior correlates of welfare are important non-invasive tools for routine management of captive animals, such as the use of quantitative observations of the spatial distribution of animals and of behavioral 'indicators' [3]. Behavior is the result of numerous extrinsic and intrinsic processes, both physical and mental, and so is sensitive to welfare state [6,7]. Additionally, behavior can be readily and non-invasively monitored and measured, often in real time, by husbandry staff with minimal resource requirements. Validated behavioral measures are powerful tools for managing and improving welfare, but one reliant on an understanding of activity patterns in a focal species and of which behaviors are effective indicators of welfare.

Amphibians are highly threatened as a group [8]. They are widely maintained in captivity for the purposes of research [9], conservation [10–12], education [13], and as pets [14], and yet suffer from negative bias in welfare science [15]. Moreover, amphibians are a diverse group with high degrees of species-level specialization [16], making it important to

understand behavioral repertoires, activity budgets, and measures of welfare for individual species [17,18]. In the handful of species where these data have been collected under captive conditions, welfare impacts of basic husbandry conditions have been identified through behavioral measures [19–24], highlighting the importance of the development of such tools. Furthermore, for animals involved in ex situ conservation, welfare may have impacts on conservation success [25].

The Lake Oku clawed frog (*Xenopus longipes*) is a small, fully aquatic anuran species occurring in Lake Oku, Mount Oku, North West Region Cameroon [26]. The species was assessed as critically endangered on the IUCN Red List of Threatened Species [27] and was subject to a mass mortality event between 2006 and 2010, the cause of which remains unknown [28,29]. A population is maintained for captive husbandry research at ZSL London Zoo [30]. This population, one of only two populations in zoos globally, has been used to develop husbandry guidelines for the species [31], to document reproductive and life history data [30,32], and for research into foraging behavior [33] and individual identification systems for the species [34]. However, basic behavioral data, including an ethogram and activity budget and identification of behavioral indicators of welfare, are still lacking for the species.

In this study, we developed an ethogram for *X. longipes*, and used this to document the diurnal activity budget for the species and to identify behavioral correlates of welfare through validation in association with handling events.

## 2. Materials and Methods

### 2.1. Study Subjects

The study sample included 24 adult wild-caught adult *X. longipes*, consisting of seven males and seventeen females, housed at ZSL London Zoo since collection from the wild in 2008. The subjects were housed in a large unit [30] containing five occupied tanks. Tank dimensions are 45 × 45 × 45 cm, with water depth 35 cm. Each tank contains several terracotta tubes, some large stones, two plants (*Microsorium pteropus*), and 5 cm aperture plastic mesh for animals to rest on. Subjects were distributed across the five tanks in mixed sex groups with at least one male per tank. There were three tanks of five individuals, one tank of six, and one tank of three. The legal acquisition, provenance, and husbandry of the animals is provided by Michaels et al. [30].

### 2.2. Ethical Approval

Ethical approval of these methods was provided by the Faculty Research and Ethics Committee at the University of Chester (1708/20/JD/BS) and full ethical review was deemed unnecessary by the Ethics Committee at ZSL as all methods fell within normal husbandry practice (ZDR435).

### 2.3. Ethogram

Observations to establish an ethogram for the study species lasted a total of three hours and were conducted live between 1030 h and 1530 h on 22 February 2021. During the continuous observation period, the observer noted all behaviors witnessed ad-lib across the subject group including males and females, and frogs in different tanks. Descriptions were provided for each behavior. All event and state behaviors were noted and adapted from previous work with closely related species, such as *Xenopus laevis* [35].

### 2.4. Baseline Behavioral Data

In order to generate an activity budget for the species, each occupied tank was recorded for a total of three hours in February 2021 using Samsung S10 HMX-H200BP, Canon Legria HFR706, and SONY DCR-SX30 camcorders. Data were collected at three times per day: a morning session (10:30 to 11:30 a.m.), a noon session (12:30 to 13:30 p.m.), and an afternoon session (14:30 to 15:30 p.m.), hereafter Time of Day, on the 22 and 26 February 2021. Nocturnal observations were not possible due to coronavirus-related limitations on staffing

and protocol, and consequential concerns regarding health and safety, despite evidence of circadian rhythms and nocturnal activity in captivity in the close relative *X. laevis* [36] and in the wild for *X. longipes* [37]. Importantly, all fundamental behaviors including reproduction are routinely observed in *X. longipes* during the day [30,31]. Cameras were positioned directly in front of the tank to be recorded to ensure maximum visibility and that the whole of each aquarium was visible. Any husbandry, including cleaning and feeding animals, was conducted after the final recording in order to avoid affecting behavior. However, small invertebrate organisms on which frogs preyed were resident in the aquarium throughout the study.

Following the completion of this filming, the BORIS software was used to record durations of swimming, resting, foraging, feeding, breathing, and amplexus behavior using the ethogram (Table 1). The use of this software allowed for continuous recording and focal sampling, as the video was repeated with the observer watching and only scoring for a different focal frog each time. This method allowed for the computation of durations of behaviors, as well as for matching of experimental data for individual frogs. Individuals were identified using belly markings and by following individuals manually through footage. An activity budget was produced to illustrate the proportion of time spent by each individual performing each behavior.

**Table 1.** Ethogram of state behaviors for captive *X. longipes*, adapted from work on *X. laevis* [35].

| Behavior | Definition |
|---|---|
| Swimming | Subject is moving from one location to another through the water, exercising front limbs, back limbs or both to travel. This may be horizontally or vertically. |
| Resting | Subject is stationary. None of the subject's limbs are being exercised to actively travel in any direction. This may be in the water or resting on a substrate. |
| Foraging | Subject is actively searching for food through a substrate using the forelimbs. This may be followed by feeding, for which a separate event is recorded. |
| Feeding | Subject is consuming a food item, rapidly wafting the item towards the face and mouth with forelimbs and often tilting body forwards following. |
| Breathing | Subject is breathing at the surface of the water with the nares breaching the surface. |
| Sloughing | Subject forces out limbs in order to removed shed skin. Swimming will likely become rapid and uncontrolled. The slough is often consumed. |
| Amplex | A male frog grips a female around her inguinal region as part of reproductive behaviour. |

The data were collected by only one observer who was trained in the software by a member of staff at the University of Chester. At the time of study, the observer was an MRes Biological Sciences student (graduated October 2022 with Merit); the observer has experience of behavioral study in a range of taxon, gained through a BSc Animal Behavior and Welfare and experience in the zoological industry. The observer has experience recording the behavior of other *Xenopus* species and was trained in doing so by members of the Amphibian Behaviour and Endocrinology Group at the University of Chester. The observer received additional training relevant to *X. longipes* on section prior to the study with staff who work with the species professionally.

Statistical Analysis

Total durations of behaviors were calculated separately for each frog for each time of day (AM, noon, PM). These data were analyzed using linear mixed models via the Lmer and lme4 packages [38,39] in R version 4.1.1 using RStudio Version 1.4.17. Model choice was informed by the Akaike Information Criterion (AIC); interactions were not included as these models resulted in an increased AIC value. Models used each behavior as a response variable, with sex, individual ID, tank number, and time of day being explanatory variables. Individual ID, nested within Tank, was a random factor to control for repeated measures and nested aspects of the design. The anova (model) function was used to test for effects of explanatory variables, and the emmeans package [40] was used for pairwise comparison when a significant effect of session (the only explanatory variable of interest with more than two levels) was detected. Swimming, Resting, Foraging, Feeding, and Breathing were included in analysis. No Other or Sloughing behavior were recorded and too few observations of male-only behavior Amplexus were recorded for meaningful analysis, so these categories were not analyzed. We confirmed that model assumptions were met through visual inspection of residuals via the ggResidpanel function in R [41].

*2.5. Behavioral Response to Stressor*

The frogs underwent a routine health check, with one tank, chosen at random, being subject to the routine procedure each day for a five-day period, until each tank had been subjected to the health check once. These health checks involved removing all of the frogs in a tank at once from the water by hand, placing them in a separate container of approximately 2.5 L of water taken from their aquarium, selecting a frog at random, catching it in a gloved hand, and visually inspecting it for 30 s. Frogs were also handled on their backs in order to be swabbed on the underbelly for routine chytrid fungus surveillance using a sterile swab. Frogs were then placed in a second identical container until all individuals in the group had been checked and the group could be returned at once to the main tank simultaneously. Health checks commenced at 10:15 a.m. so that frogs were returned to the home tank and observed at a similar time to the behavioral observation sessions (time of day (AM)). All frogs were returned to the tank at the same time. Observations began immediately upon return to the tank. Observations lasted an hour, and the video cameras were set up in the same manner as for experiment two. Humans were not present for the duration of the recording sessions.

The footage was analyzed in the same way using the BORIS software, ethogram, and individual IDs to allow for pairing of data in the control and in the health check. The use of this software facilitated continuous recording and focal sampling. The data was collected by the same observer.

Statistical Analysis

Behavioral data matched for time of day (i.e., time of day (AM) data) were used to test for effects of health check on behavior. Data were analyzed for baseline data using linear mixed models via the lmer and lme4 packages in R [38,39]. Model choice was informed by the Akaike Information Criterion (AIC); interactions were not included as these models resulted in an increased AIC value. Models used each behavior as a response variable, with health check status (yes or no), sex, and individual ID nested within a tank number being explanatory variables. Individual ID, nested within a tank, was a random factor to control for repeated measures and nested aspects of the design. The anova(model) function was used to test for effects of explanatory variables, and the emmeans package [40] was used for pairwise comparison when a significant effect of session (the only explanatory variable of interest with more than two levels) was detected. Swimming, Resting, Foraging, Feeding, and Breathing were included in analysis, but No Other or Sloughing behavior were recorded and too few observations of male-only behavior Amplexus were recorded for meaningful analysis, so these categories were not analyzed. We confirmed that model

assumptions were met through visual inspection of residuals via the ggResidpanel function in R [41] for baseline data.

### 3. Results

#### 3.1. Ethogram

An ethogram was produced to identify the state behaviors exhibited by *X. longipes* in captivity (Table 1).

#### 3.2. Baseline Behavioral Data

Sloughing was so rarely observed that although it was recorded once during the ethogram construction, it was not observed at all during subsequent observations. Linear mixed models, with sex, individual ID, tank number, and time of day being explanatory variables, showed that there was no effect of frog sex or time of observation on any behavior other than an effect of sex on duration of breathing (Table 2). An activity budget pooled across all sessions and both sexes is presented in Figure 1. Parameter estimates of the models are presented in Table 4. Sloughing and Amplexus behavior were almost never recorded, and data were not analyzed. Amplexus accounted for 1.3% of total budget and was only observed three times across all frogs and all observations; it could also only be exhibited by males.

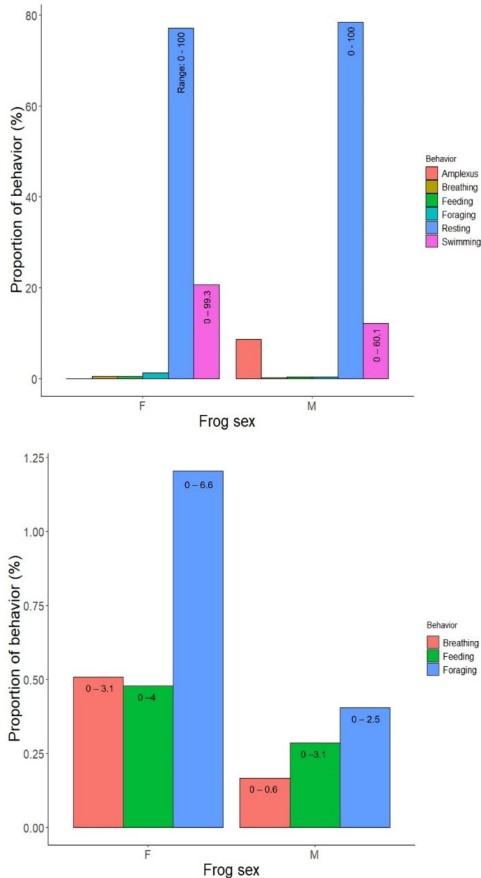

**Figure 1.** Diurnal activity budget of male and female *X. longipes*; data are expressed as percentages of total behavior duration pre-disturbance across AM, noon, and PM observations. Sloughing is not shown as it was not recorded during the observation periods. The lower pane shows the three behaviors that are not visible in the top pane at a different scale for clarity. The range of percentages has been displayed for each behavior by subtracting the smallest percentage of total behavior predisturbance from the greatest percentage of total behavior pre-disturbance.

**Table 2.** Results of linear mixed models with sex and time of observation as explanatory variables. Significant *p* values are in bold.

| Behavior | Effect of Sex | Effect of Session |
|---|---|---|
| Swimming | $F_{1,22} = 2.54$, $p = 0.13$ | $F_{2,46} = 3.14$, $p = 0.053$ |
| Resting | $F_{1,22} = 0.16$, $p = 0.69$ | $F_{2,46} = 0.73$, $p = 0.49$ |
| Foraging | $F_{1,22} = 4.0$, $p = 0.06$ | $F_{2,46} = 0.02$, $p = 0.98$ |
| Feeding | $F_{1,22} = 0.79$, $p = 0.38$ | $F_{2,46} = 1.5$, $p = 0.2286$ |
| Breathing | $F_{1,22.003} = 5.36$, $\boldsymbol{p = 0.03}$ | $F_{2,46.003} = 0.53$, $p = 0.59$ |

*3.3. Behavioral Response to Stressor*

All behaviors measured were significantly affected by the health check (Table 3; Figure 2), but no effect of sex was detected (Table 3). The proportion of time spent exhibiting Swimming and Feeding behaviors increased, whilst Resting, Foraging, and Breathing behaviors decreased. Parameter estimates of the models are presented in Table 3. Sloughing and Amplexus behavior data were not analyzed as the former was not recorded in main observation sessions and the latter was too rarely detected to yield data for meaningful analysis.

**Table 3.** Results of linear mixed models with health check and sex as explanatory variables. Significant *p* values are in bold.

| Behavior | Effect of Health Check | Effect of Sex |
|---|---|---|
| Swimming | $F_{1,22.58} = 171.5$, $p < \boldsymbol{0.001}$ | $F_{1,34.3} = 1.755$, $p = 0.19$ |
| Resting | $F_{1,22.58} = 171.5$, $p < \boldsymbol{0.001}$ | $F_{1,34.3} = 1.756$, $p = 0.19$ |
| Foraging | $F_{1,45} = 6.2$, $p = \boldsymbol{0.016}$ | $F_{1,45} = 2.38$, $p = 0.13$ |
| Feeding | $F_{1,45} = 7.46$, $p = \boldsymbol{0.009}$ | $F_{1,45} = 1.40$, $p = 0.24$ |
| Breathing | $F_{1,22.9} = 12.62$, $p = \boldsymbol{0.002}$ | $F_{1,28.3} = 3.06$ $p = 0.09$ |

**Table 4.** Effect parameters from linear mixed models of baseline behaviors, as a factor of sex and time of day, and of behaviors as a factor of sex and health check.

| Model | Response Variable | Parameter | Estimate (SD for Random Effect) | Standard Error of Estimate | *t* Value | Lower 95% CI of Estimate | Upper 95% CI of Estimate |
|---|---|---|---|---|---|---|---|
| Behavior = sex + time of day + frog (tank) | Swimming | Intercept | 656.43 | 159.51 | 4.115 | 348.306 | 964.553 |
| | | Sex (M) | −399.70 | 250.75 | −1.594 | −889.705 | 90.303 |
| | | Time (noon) | 174.00 | 145.98 | 1.192 | −111.794 | 459.794 |
| | | Time (pm) | 365.70 | 145.98 | 2.505 | 79.906 | 651.494 |
| | | R² Marginal | 0.104 | - | - | - | - |
| | | R² Conditional | 0.525 | - | - | - | - |
| | | Random effect | 475.9 | - | - | - | - |
| | Resting | Intercept | 2781.37 | 200.78 | 13.853 | 2392.367 | 3170.374 |
| | | Sex (M) | 136.04 | 335.39 | 0.406 | −519.360 | 791.448 |
| | | Time (noon) | −99.15 | 150.05 | −0.661 | −392.906 | 194.606 |
| | | Time (pm) | −180.75 | 150.05 | −1.205 | −474.506 | 113.006 |
| | | R² Marginal | 0.013 | - | - | - | - |
| | | R² Conditional | 0.638 | - | - | - | - |
| | | Random effect | 683.9 | - | - | - | - |

**Table 4.** *Cont.*

| Model | Response Variable | Parameter | Estimate (SD for Random Effect) | Standard Error of Estimate | *t* Value | Lower 95% CI of Estimate | Upper 95% CI of Estimate |
|---|---|---|---|---|---|---|---|
| | | Intercept | 39.59 | 10.96 | 3.611 | 18.463 | 60.725 |
| | | Sex (M) | −26.72 | 13.36 | −2.000 | −52.830 | 0.6149 |
| | | Time (noon) | 3.00 | 14.30 | 0.210 | −24.934 | 30.934 |
| | Foraging | Time (pm) | 2.10 | 14.30 | 0.147 | −25.834 | 30.034 |
| | | $R^2$ Marginal | 0.057 | - | - | - | - |
| | | $R^2$ Conditional | 0.082 | - | - | - | - |
| | | Random effect | 8.188 | - | - | - | - |
| | | Intercept | 23.609 | 5.718 | 4.129 | 12.592 | 34.625 |
| | | Sex (M) | −6.373 | 7.184 | −0.887 | −20.412 | 7.666 |
| | | Time (noon) | −8.400 | 7.275 | −1.155 | −22.642 | 5.842 |
| | Feeding | Time (pm) | −12.450 | 7.275 | −1.711 | −26.692 | 1.792 |
| | | $R^2$ Marginal | 0.050 | - | - | - | - |
| | | $R^2$ Conditional | 0.112 | - | - | - | - |
| | | Random effect | 6.649 | - | - | - | - |
| | | Intercept | $1.930 \times e^{01}$ | 4.205 | 4.589 | 11.186 | 27.408 |
| | | Sex (M) | $-1.165 \times e^{01}$ | 5.031 | −2.315 | −21.308 | 1.986 |
| | | Time (noon) | $-4.174 \times e^{-14}$ | 5.558 | 0.00 | −10.757 | 10.757 |
| | Breathing | Time (pm) | −4.950 | 5.558 | −0.891 | 15.707 | 5.807 |
| | | $R^2$ Marginal | 0.083 | - | - | - | - |
| | | $R^2$ Conditional | 0.088 | - | - | - | - |
| | | Random effect | 1.395 | - | - | - | - |
| | | Intercept | 608.284 | 114.36 | 5.319 | 384.294 | 829.647 |
| | | Healthcheck (yes) | 1397.250 | 106.70 | 13.095 | 1184.948 | 1609.553 |
| | Swimming | Sex (M) | −234.632 | 177.11 | −1.325 | −578.974 | 119.898 |
| | | $R^2$ Marginal | 0.671 | - | - | - | - |
| | | $R^2$ Conditional | 0.820 | - | - | - | - |
| | | Random effect | 336.5 | - | - | - | - |
| Behavior = health check status + sex + frog (tank) | | Intercept | 608.284 | 160.25 | 17.547 | 384.294 | 829.647 |
| | | Health check (yes) | 1397.250 | 183.11 | −6.839 | 1184.948 | 1609.553 |
| | Resting | Sex (M) | −234.632 | 239.44 | 0.132 | −578.974 | 119.898 |
| | | $R^2$ Marginal | 0.671 | - | - | - | - |
| | | $R^2$ Conditional | 0.820 | - | - | - | - |
| | | Random effect | 336.5 | - | - | - | - |
| | | Intercept | 37.187 | 8.444 | 4.404 | 20.836 | 53.537 |
| | | Health check (yes) | −27.150 | 10.875 | −2.497 | −48.206 | −6.093 |
| | Foraging | Sex (M) | −18.469 | 11.962 | −1.544 | −41.632 | 4.694 |
| | | $R^2$ Marginal | 0.155 | - | - | - | - |
| | | $R^2$ Conditional | 0.155 | - | - | - | - |
| | | Random effect | 0.00 | - | - | - | - |

**Table 4.** *Cont*.

| Model | Response Variable | Parameter | Estimate (SD for Random Effect) | Standard Error of Estimate | *t* Value | Lower 95% CI of Estimate | Upper 95% CI of Estimate |
|---|---|---|---|---|---|---|---|
| Feeding | | Intercept | 30.251 | 17.36 | 1.743 | −3.589 | 63.861 |
| | | Health check (yes) | 61.050 | 22.35 | 2.731 | 17.765 | 104.334 |
| | | Sex (M) | −29.148 | 24.59 | −1.185 | −76.762 | 18.467 |
| | | $R^2$ Marginal | 0.159 | - | - | - | - |
| | | $R^2$ Conditional | 0.159 | - | - | - | - |
| | | Random effect | 0.00 | - | - | - | - |
| Breathing | | Intercept | 18.164 | 3.025 | 6.004 | 12.307 | 24.024 |
| | | Health check (yes) | −12.900 | 4.630 | −3.553 | −20.137 | −5.663 |
| | | Sex (M) | −7.762 | 4.439 | −1.749 | −16.541 | 0.826 |
| | | $R^2$ Marginal | 0.235 | - | - | - | - |
| | | $R^2$ Conditional | 0.326 | - | - | - | - |
| | | Random effect | 4.612 | - | - | - | - |

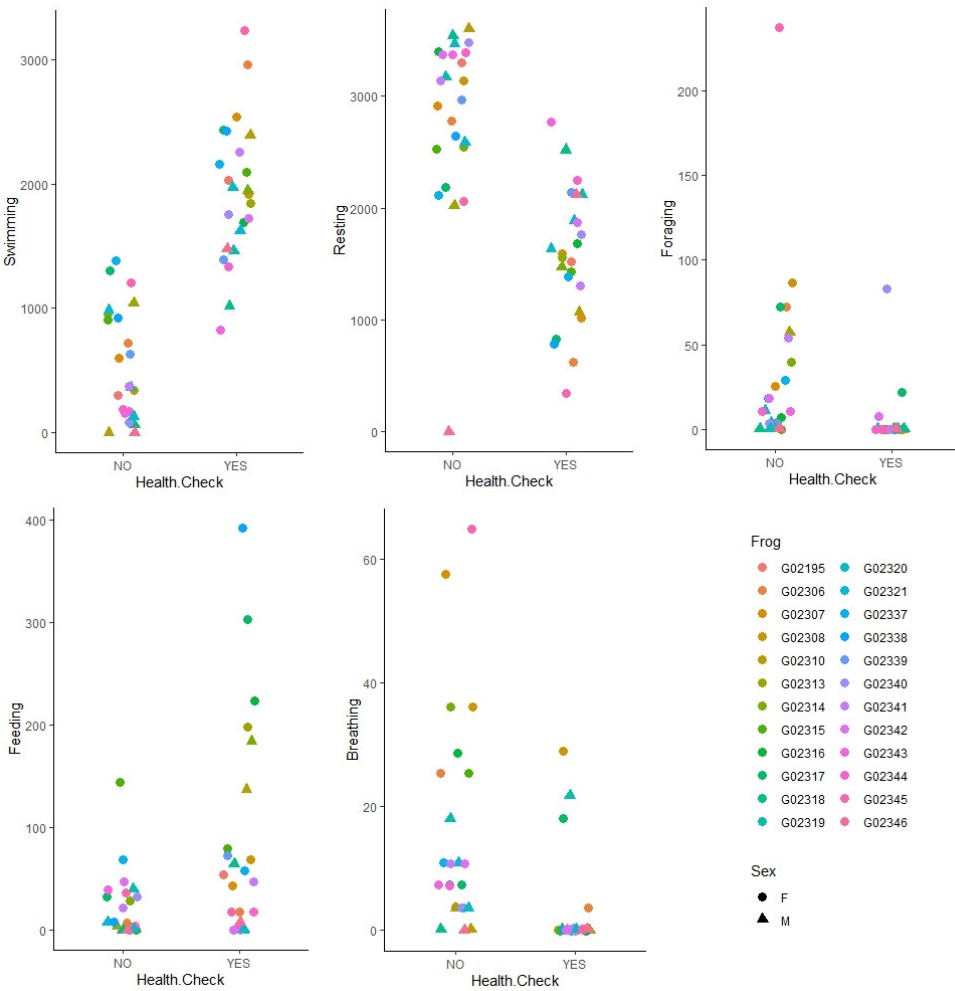

**Figure 2.** Mean durations of behaviors under baseline and post-health check conditions in *X. longipes*. There was a significant effect of health check on each behavior (see Table 3). Sloughing and Amplexus behavior data were not analyzed as the former was not recorded in main observation sessions and the latter was too rarely detected to yield enough data for analysis.

## 4. Discussion

We created the first ethogram for *X. longipes*, describing swimming, resting, foraging, feeding, breathing, amplexus, and sloughing behaviors observed by captive *X. longipes*. Although the behavioral repertoire seen in the ethogram is limited in comparison to that of *X. laevis* [35], it is more comprehensive than ethograms available for other amphibian species [42].

Behaviors which have been previously identified as potential stress indicators in other *Xenopus* species were not observed in *X. longipes*. For instance, walling behavior in *X. laevis* was previously described [21] as "Fast swimming back and forwards along a tank wall; rapid rear limb kicks; scrabbling at tank walls with forelimbs; snout against tank wall". Whilst swimming behaviors were detected in this species, the threshold for walling behavior could not be met as "rapid rear limb kicks", "scrabbling at tank walls with forelimbs", and "snout against tank wall" was not present during the observation period (JED, personal observation). Furthermore, although the speed of swimming may have increased in some instances in this study, it was not quantified to identify as "fast swimming".

Little is known about the biology of *X. longipes* [30,43]. The analogue species concept is widely used in the development of amphibian conservation breeding programs [16,44] whereby common relatives of a threatened species are used as models to develop husbandry protocols prior to working with target species [45]. Previously published studies have demonstrated the limitations of the analogue species concept with regard to assumptions made regarding reproductive biology and larval development [30,46]. This study could indicate further limitations of this concept when comparing behaviors of congeneric species. The production of this ethogram highlights the importance of species-specific behavior and welfare research and the caution which should be taken when comparing the behavior of species, even within the same genus. Therefore, husbandry and care practices should be reflective of species-specific natural behavioral biology.

There was no significant difference in any behavior across the sessions at three different times of the day. However, there is evidence of increased nocturnal locomotor activity in *X. laevis* [36], which could also be the case for *X. longipes*. Therefore, a comparison of diurnal and nocturnal behavior may yield significant differences; this was outside the scope of this study. As a result, future applications of this work may not have to control for the time of day. Although nocturnal observation would be useful to inform baseline activity budgets in this study, this was impossible within resource constraints as similar video cameras equipped with infrared night-vision simply created a glare from the glass that prevent ed observation. *X. longipes* is relatively diurnal compared with *X. laevis*; although greater shoreline activity is noted at night in the field [37], captive animals routinely exhibit all fundamental behaviors including locomotion, feeding, and reproduction during the day [30,31], and the data presented here demonstrate that a range of behaviors was detected. From the perspective of practical application, husbandry interventions causing stress, and keeper observations to quantify welfare, all take place during the day, so diurnal behavioral patterns are most relevant. Future work should include nocturnal data collection.

A significant difference exists in breathing duration between the sexes in *X. longipes* where females spend more time breathing than males. Given the much smaller size of males than females [43], this may be the result of differing volume:surface area ratios and implications thereof on the proportion of gas exchange requirements that can be met through cutaneous routes. However, our data do not allow for a clear reason to be identified and other mechanisms may exist. Consequently, differences in behavior between the sexes should still be considered in future work regarding this species.

Our data show that these frogs spend the vast majority of their time swimming and resting, with little of their activity budget allocated to other behaviors. The proportion of time spent swimming was broadly similar (between 10 and 20% of total time) to that reported for *X. laevis* previously [20 (under the condition where refuge was present in this experiment), 21]. Comparisons for other behaviors are not available in the literature. This species does engage in complex feeding behavior when food is present [33], and it

is important to note that the present behavioral budget is specifically for frogs outside of when food is delivered to systems; a predominance of foraging and feeding behavior would be expected at these points. Although the breeding season for this species has not yet been identified and in captivity it appears to be sporadic and linked to favorable environmental parameters [31], Amplexus was relatively rarely observed, both in terms of duration (Figure 1), but especially in terms of number of bouts (only three across all observations). We recognize that during breeding periods this may increase substantially. Additionally, our data derive from groups of frogs, which will inevitably perturb individual behavior through interactions between conspecifics. However, given that this species is routinely kept in groups in captivity [30] and observed in groups in close proximity to one another in the field [44], we believe that our data are a good representation of the norm for this species.

The models used for the baseline data have relatively low marginal and (other than for Swimming and Resting) conditional $R^2$ values and relatively broad confidence intervals around parameter estimates, indicating a large amount of variation in behavior durations, and supportive of sex and time of day explaining little variation. For Swimming, Resting, and Feeding, the conditional $R^2$ is much higher than the marginal, and for the former two in this list, these values are close to one. Standard deviations of the random effect are also reasonably high. This suggests that in these models, frog identity (nested within tank) explained a substantial amount of variation, and that there may be consistency between individuals in the durations of these behaviors that is not linked to their sex.

Swimming, resting, foraging, feeding, and breathing behaviors were all significantly affected by the health check (Figure 2). A change in behavior was seen in *X. laevis* when subjected to unnatural environmental conditions and was linked with an increase in corticosterone [21]. One explanation for the increase in swimming following the health check in *X. longipes* could be the presence of an escape response which likely mirrors the increase in walling behavior in *X. laevis* during the stress response. Although we did not identify 'walling' behavior [21] as a qualitatively separate behavior from Swimming in our study, increased Swimming could be compared to the increase in walling seen in stressed *X. laevis*; the relatively small physical size of *X. longipes* individuals relative to tank size may have reduced boundary interaction, which is part of the definition of walling. Whilst walling may have been observed over a longer observation period, the 1 h period after a health check was selected as previous work on *X. laevis* has recorded walling behavior within half an hour of experiencing a stressor [21]. Furthermore, anecdotally, walling has not been reported by keeping staff in this species. It seems likely that Swimming behavior induced by stress in *X. laevis* becomes walling once animals interact with a transparent barrier, while *X. longipes* follows the barrier but does not react by swimming up the barrier.

The increase seen in Feeding behavior is likely the result of frogs encountering potential food items in the aquarium more frequently due to the increase in Swimming behavior. The models used for the health check data have moderate to high marginal and conditional $R^2$ values, indicating that these models are a good fit, and that there is a relatively strong effect of health check despite substantial variation in the data (Figure 2. For two behaviors (Foraging and Feeding), the random effect standard deviation is zero (the *lme4* package reports outcomes of zero when the value is very close to zero), indicating that differences between individuals that cannot be explained by the rest of the model are negligible in this case.

One notable difference in the experimental design of this study and investigations applying welfare assessment tools to *X. laevis* [21,47,48] is that frogs in previous works have been separated into individual tanks for the observation periods. As the subjects are usually kept in groups [31], separation was deemed to be an unnecessary cause of stress in this study. Nonetheless, although data analysis controlled for tank, it is possible that behavior was influenced by interactions between individuals within a tank. This interaction is relevant to the practical application of the data, however, as this species is usually kept in a group.

Although the behavioral changes we detected, given the context, are strongly suggestive of a stress response and align with research in congeners [20,21], validation of this would require that behavior be correlated with corticosterone levels [43]. However, the methods used to quantify corticosterone release rates for *X. laevis* have not yet been validated for use in *X. longipes.* In order to do so, rigorous validation experiments would be required to undergo technical validation, to confirm the sensitivity, specificity, accuracy, and precision of the assay, and biological validation before application to this species [44]; this was outside the scope of this study. In the interim, we suggest that behavioral changes shown here may be used as an indicator of probable stress response to at least short-term disturbance, which may be used to inform husbandry practices. Our results indicate a behavioral impact of the capture of frogs for veterinary monitoring, consistent with a stress response in this genus [20,21], with respect to duration of swimming behavior and repetitive swim patterns. These data highlight the importance of tempering the need to monitor the health of captive animals with the impact of doing so on their welfare and emphasize the need to use non-invasive methods to monitor animals where possible.

There are minimal studies regarding the impact of health checks on amphibian species. Capture, restraint, and handling has been used in the biological validation of corticosterone detection methods for *Ambystoma andersoni* [49] which elicited an increase in corticosterone release, inactivity and gill beat rates following a health check. The contrasting increase in inactivity in *A. andersoni* and increase in activity in *X. longipes* further highlights the need for species-specific research in this area.

This study provides a strong foundation for further research on *X. longipes*, following models used for other pipid taxa to optimize husbandry [20,21,50]. Using the behaviors identified in the ethogram and as potential indicators of stress, investigations can begin assessing husbandry and housing conditions for the species in captivity in order to enhance conservation goals. Investigations into husbandry practices, and other welfare related questions, could be confirmed with the use of corticosterone analysis. If the methods used to quantify corticosterone release rates available for *X. laevis* can be validated for *X. longipes*, further investigation could confirm the potential for increased swimming as an indicator of stress. If a significant rise in the behavior correlates with greater corticosterone release rates, the potential for this behavior as a non-invasive welfare assessment tool can be established [21].

## 5. Conclusions

This investigation has produced a detailed ethogram for *X. longipes*, establishing six recognizable and observable behaviors. These behaviors can now be applied to further research into husbandry and management practices for the species. Application of these findings may enhance conservation and animal welfare goals by providing evidence needed to better evaluate captive husbandry protocols.

Comparison of behavior in the control and following the health check revealed a significant difference in many behaviors, including increases in Swimming and Feeding alongside decreases in Resting, Foraging, and Breathing. An increase in swimming was linked to walling behavior, although not all aspects of walling were observed. Swimming also became more repetitive, which was illustrated by the decrease in Breathing and Foraging. Increased swimming duration and repetitiveness could be a potential stress-indicator behavior for the species, although this should be confirmed with corticosterone analysis.

Corticosterone analysis could be used to further investigate the duration of the stress response if methods used to quantify corticosterone release rates in *X. laevis* can be applied to *X. longipes.* This would confirm the potential for these behaviors as non-invasive indicators of welfare for this species in captivity.

**Author Contributions:** Conceptualization, J.E.D., C.J.M., C.E., T.E.S. and C.A.H.; methodology, J.E.D. and C.J.M.; software, J.E.D. and C.J.M.; validation, J.E.D. and C.J.M.; formal analysis, J.E.D. and C.J.M.; investigation, J.E.D. and C.J.M.; resources, J.E.D., C.J.M., C.E. and B.T.; data curation, J.E.D., C.J.M. and C.E.; writing—original draft preparation, J.E.D. and C.J.M.; writing—review and editing, J.E.D., C.J.M., B.T., T.E.S. and C.A.H.; visualization, J.E.D. and C.J.M.; supervision, C.J.M., T.E.S. and C.A.H.; project administration, J.E.D. and C.J.M.; funding acquisition, J.E.D., T.E.S. and C.A.H. All authors have read and agreed to the published version of the manuscript.

**Funding:** This research was funded by a small student bursary grant from the University of Chester, covering transport and equipment.

**Institutional Review Board Statement:** Ethical review and approval were waived for this study as it did not deviate from normal husbandry practice. Please see Methods for details of institutional review and approval.

**Informed Consent Statement:** Not applicable.

**Data Availability Statement:** Data are available https://github.com/CJMichaels/Dias-et-al-Xenopus-longipes-health-check-welfare (accessed on 15 April 2022).

**Acknowledgments:** The authors would like to thank Francesca Servini, Daniel Kane and Unnar Aevarsson at ZSL London Zoo for supporting the practical elements of the study, carrying out husbandry on the animals and Daniel Kane for comments on an earlier version of the manuscript. The authors would also like to thank staff within the Biological Sciences department at the University of Chester, including Michal Zatrak who helped to fine tune the experimental design. The captive colony of *X. longipes* was exported in 2008 and 2012 under permit from the Cameroon Ministry of Forestry & Wildlife (0928/PRBS/MINFOF/SG/DFAP/SDVEF/SC and 0193/CO/MINFOF/SG/DFAP/SDVEF/SC), following prior consultation with the Oku community.

**Conflicts of Interest:** The authors declare no conflict of interest.

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
