# Peer review of "Baseline Behavioral Data and Behavioral Correlates of Disturbance for the Lake Oku Clawed Frog (Xenopus longipes)"

_2673-5636, doi:10.3390/jzbg3020016_

Round 1
Reviewer 1 Report
It's fantastic to see an example of useful information for understudied species which holders can use to monitor welfare on their own animals.
Only small concerns for this well presented paper:
Line 17 – Re-word for clarity “…measure the behavioural impact of a routine health-check where frogs were restrained”
Line 224 – End of sentence missing?
On reading the methods and discussion I felt that it was a relatively short data collection period (3 x 1 hour over 2 days). Could this also be an explanation of why walling behaviour was not observed or is it unlikely for the species? Perhaps add a sentence to the discussion as a possible limitation with a small data set and boost this with any anecdotal evidence from keeping staff if it’s felt that walling behaviour is less likely to occur in X. longipes?
Author Response
We are grateful for the comments provided by the refere and have addressed each in turn, below.
Reviewer 1 Report:
Comments and Suggestions for Authors: It's fantastic to see an example of useful information for understudied species which holders can use to monitor welfare on their own animals. Only small concerns for this well presented paper.
Author Response: The authors appreciate the reviewer's recognition of the value of this work and its potential applications to animal welfare monitoring for the study species.
Line 17 – Re-word for clarity “…measure the behavioural impact of a routine health-check where frogs were restrained”
Author Response: This sentence has been adjusted.
Line 224 – End of sentence missing?
Author Response: This sentence has been revised and completed.
On reading the methods and discussion I felt that it was a relatively short data collection period (3 x 1 hour over 2 days). Could this also be an explanation of why walling behaviour was not observed or is it unlikely for the species? Perhaps add a sentence to the discussion as a possible limitation with a small data set and boost this with any anecdotal evidence from keeping staff if it’s felt that walling behaviour is less likely to occur in X. longipes?
Author Response: A sentence has been added to recognise the limitations of this short observation period and to justify why this length of time had been selected. Furthermore, a sentence has been added to confirm that walling behavior has not been reported by keeping staff in this species. Some more detail on the potential development of walling behaviour has also been added.
Reviewer 2 Report
This straightforward project is well designed and explained. I appreciate getting basic behavioral information out for an amphibian species. The text as written has a couple of small things that require a quick look, mostly grammar.
Abstract, line 17. Remove hyphen in “health-check”
Intro, line 33. Need a hyphen between "species" and "specific"
Intro, line 40-41. There's a section of this sentence that is unclear: "Once validated, measuring behavior..." to "... and improving welfare" should be revised. I suggest "Validated behavioral measures are powerful tools for managing and improving welfare, but they are reliant..."
Materials and methods, line 72. Can you share the tank sizes or maybe even include a picture or diagram?
Materials and methods, line 96. Were night observations not possible due to inability to find options for nocturnal illumination?
Materials and methods, line 99. Add "that" between "...visibility and..." and "...the whole..."
Materials and methods, line 103. Change "these observations" to "this filming"
Materials and methods, line 105-107. I'm a little confused by this sentence. Presumably the "this" on line 105 is the software? If so, try "This software..." Your actual observation method is not completely clear. It sounds like you are doing continuous observations (so you can computer durations) and going through the video multiple times, watching and only scoring for a different focal frog each time. If that's correct, try to make that method more explicit by revising this sentence.
Materials and methods, line 111. There's was/were confusion in this line. My suggestion is "Total durations of behaviors were calculated separately for each frog for..."
Materials and methods, line 119. Parenthesis error--it's duplicated after "interest" and after "levels". I suggest removing it after "interest."
Materials and methods, line 120. Add a period after "analysis." And drop the but to start the next sentence "No Other or Sloughing behaviors were recorded, ..."
Materials and methods, line 121-122. Try "... too few observations of male-only behavior Amplexus were recorded for meaningful analysis, so these categories..."
Materials and methods, line 126. How was the tank chosen? Random selection? Was it the same one for all five days, or did tank used vary daily?
Materials and methods, line 152-154. See previous notes on 118-121.
Table 1. I'm assuming all behaviors are states and not instantaneous? That information should be explicitly included.
Results, line 166. Did you look at interaction effects?
Results, line 183-187. I realize you've got some of this information in the tables, but you should explain in text as the information about the direction of the changes.
Table 4. I feel that table 4 can be removed in its entirety.
Discussion, line 224. This sentence just ends partway through! I think it's missing "differences" from the end. I.e., "...should be reflective of species-specific behavioral differences."
Discussion, line 246. What is the season for this species? Is it affected by captivity?
Discussion, line 269-270. Again, what are tank dimensions?
Discussion, line 309. Weird spacing between "with" and "greater"
Author Response
We are grateful for the comments provided by the refere and have addressed each in turn, below.
This straightforward project is well designed and explained. I appreciate getting basic behavioral information out for an amphibian species. The text as written has a couple of small things that require a quick look, mostly grammar.
Author Response: The authors appreciate the reviewer’s acknowledgment of the strengths of this experimental design. Particularly as this was a Masters’ by Research level project conducted by the first author.
Abstract, line 17. Remove hyphen in “health-check”
Author Response: This hyphen has been removed.
Intro, line 33. Need a hyphen between "species" and "specific"
Author Response: This hyphen has been added.
Intro, line 40-41. There's a section of this sentence that is unclear: "Once validated, measuring behavior..." to "... and improving welfare" should be revised. I suggest "Validated behavioral measures are powerful tools for managing and improving welfare, but they are reliant..."
Author Response: This sentence has been adjusted.
Materials and methods, line 72. Can you share the tank sizes or maybe even include a picture or diagram?
Author Response: Details on tank size have been added. The tanks were 45 x 45 x 45 cm in dimension and water depth was 35 cm.
Materials and methods, line 96. Were night observations not possible due to inability to find options for nocturnal illumination?
Author Response: Nocturnal observations were not possible due to coronavirus related restrictions and health and safety protocols. However, more detail has been added in the discussion to address the limitations of nocturnal observations in this case, had these observations been possible.
Materials and methods, line 99. Add "that" between "...visibility and..." and "...the whole..."
Author Response: Wording has been adjusted.
Materials and methods, line 103. Change "these observations" to "this filming"
Author Response: Wording has been adjusted.
Materials and methods, line 105-107. I'm a little confused by this sentence. Presumably the "this" on line 105 is the software? If so, try "This software..." Your actual observation method is not completely clear. It sounds like you are doing continuous observations (so you can computer durations) and going through the video multiple times, watching and only scoring for a different focal frog each time. If that's correct, try to make that method more explicit by revising this sentence.
Author Response: Wording has been adjusted. The paragraph has been revised to provide more detail regarding the continuous nature of observations, the focal sampling and the repeats of the videos.
Materials and methods, line 111. There's was/were confusion in this line. My suggestion is "Total durations of behaviors were calculated separately for each frog for..."
Author Response: The sentence has been adjusted.
Materials and methods, line 119. Parenthesis error--it's duplicated after "interest" and after "levels". I suggest removing it after "interest."
Author Response: Parenthesis removed.
Materials and methods, line 120. Add a period after "analysis." And drop the but to start the next sentence "No Other or Sloughing behaviors were recorded, ..."
Author Response: The sentence has been adjusted.
Materials and methods, line 121-122. Try "... too few observations of male-only behavior Amplexus were recorded for meaningful analysis, so these categories..."
Author Response: Wording has been adjusted.
Materials and methods, line 126. How was the tank chosen? Random selection? Was it the same one for all five days, or did tank used vary daily?
Author Response: The sentence has been adjusted to clarify the random nature of tank selection and that each tank was subjected to the health check only once.
Materials and methods, line 152-154. See previous notes on 118-121.
Author Response: The sentence has been adjusted.
Table 1. I'm assuming all behaviors are states and not instantaneous? That information should be explicitly included.
Author Response: Clarification has been provided regarding the nature of the state behaviors described in the ethogram.
Results, line 166. Did you look at interaction effects?
Author Response: Model choice was informed by the Akaike Information Criterion (AIC); interactions were not included as these models resulted in an increased AIC value.
Results, line 183-187. I realize you've got some of this information in the tables, but you should explain in text as the information about the direction of the changes.
Author Response: Detail added about the direction of the changes (Swimming and Feeding increased; Resting, Foraging and Breathing decreased).
Table 4. I feel that table 4 can be removed in its entirety.
Author Response: Presenting these data is important for a full understanding of the models we ran.
Discussion, line 224. This sentence just ends partway through! I think it's missing "differences" from the end. I.e., "...should be reflective of species-specific behavioral differences."
Author Response: The sentence has been revised and completed.
Discussion, line 246. What is the season for this species? Is it affected by captivity?
Author Response: Detail provided to highlight that the season for this species has not been described.
Discussion, line 269-270. Again, what are tank dimensions?
Author Response: Details on tank size have been added.
Discussion, line 309. Weird spacing between "with" and "greater"
Author Response: Spacing has been adjusted.
Reviewer 3 Report
In this study, the authors produced an ethogram for Xenopus longipes. Comparison of behavior in the control and following the health check revealed a significant difference in some behaviors. Application of these findings may enhance conservation and animal welfare goals. The authors need to notice and properly reply the following comments.
Compulsory Revisions:
- Clawed frogs should be nocturnal and most behaviors including reproductive activity and feeding occur after dark. However, nocturnal observation was not made in this study and it would be a main defect or incompleteness causing bias in quantifying the frequencies of specific types of behaviors. Note that nocturnal observations are feasible by using camcorders with IR Night Vision.
- The behavioral data of Xenopus longipes are not fully investigated in this study. What are the behavioral data when different food types are delivered to systems? What behavioral data of this study can be explicitly applied to future research regarding husbandry and management practices?
Small Revisions:
- At line 74: “5cm” should be “5 cm”.
- At line 105: “table 1” should be “Table 1”.
- At lines 117, 150: “eemeans” should be “emmeans”.
- At lines 110-124, 142-156: the two paragraphs for Statistical Analysis should be consolidated into one paragraph.
- At line 166: “table 2” should be “Table 2”.
- At line 167: “figure 1” should be “Figure 1”.
- At line 168: “Table 4” should not be presented here if “Table 3” has not been presented first.
- At line 177: how to calculate the range of percentages?
- At line 184: “figure 2” should be “Figure 2”.
- At Table 4: all “e+00” are redundant. The t values for Swimming in the second model are missing.
- At line 224: the sentence is not adequately ended.
- At line 230: “X” should be “X.”.
- At lines 273-274: I don’t agree that the models used for the baseline data have ‘moderate to high’ marginal and conditional values.
- At line 411: “Afr. J. Herpetol.” should be italicized.
- At lines 460-463: the two references are not cited in the text?
Author Response
We are grateful for the comments provided by the refere and have addressed each in turn, below.
Compulsory Revisions:
Clawed frogs should be nocturnal and most behaviors including reproductive activity and feeding occur after dark. However, nocturnal observation was not made in this study and it would be a main defect or incompleteness causing bias in quantifying the frequencies of specific types of behaviors. Note that nocturnal observations are feasible by using camcorders with IR Night Vision.
Author Response: Nocturnal observations were not possible due to coronavirus related restrictions and health and safety protocols. However, more detail has been added in the discussion to address the limitations of nocturnal observations in this case, had these observations been possible.
The behavioral data of Xenopus longipes are not fully investigated in this study. What are the behavioral data when different food types are delivered to systems?
Author Response: In this study, the authors wanted to provide a fundamental behavioral foundation for this species for the first time. Therefore, factors which may influence behavior outside of the health check needed to be controlled. This includes food delivery during observation alongside other factors such as the presence of visitors or changes in the weather. Future research investigating the husbandry practices used for the species may want to explore the influence of different food types on the behaviors described here.
What behavioral data of this study can be explicitly applied to future research regarding husbandry and management practices?
The behavioral data which can be applied to future research includes the ethogram, and the increase in swimming behavior as a potential indicator of stress. Furthermore, data on the lack of effect of time of day suggests that this factor does not need to be controlled for in future research. However, limited differences in behavior between the sexes mean that this factor should still be considered in further research. Details have been added to the discussion clarify this.
Small Revisions:
At line 74: “5cm” should be “5 cm”.
Author Response: Spacing has been adjusted.
At line 105: “table 1” should be “Table 1”.
Author Response: Capitalisation has been adjusted.
At lines 117, 150: “eemeans” should be “emmeans”.
Author Response: Spelling has been adjusted
At lines 110-124, 142-156: the two paragraphs for Statistical Analysis should be consolidated into one paragraph.
Author Response: The two paragraphs have been kept separate so that it is clear which analysis leads to which results.
At line 166: “table 2” should be “Table 2”.
Author Response: Capitalisation has been adjusted.
At line 167: “figure 1” should be “Figure 1”.
Author Response: Capitalisation has been adjusted.
At line 168: “Table 4” should not be presented here if “Table 3” has not been presented first.
Author Response: Table 4 contains parameter estimates for both models. It does not make sense to split the table as this would create too many tables to follow.
At line 177: how to calculate the range of percentages?
Author Response: Details have been provided to describe how the range of percentages were calculated.
At line 184: “figure 2” should be “Figure 2”.
Author Response: Capitalisation has been adjusted.
At Table 4: all “e+00” are redundant. The t values for Swimming in the second model are missing.
Author Response: This mistake has been addressed.
At line 224: the sentence is not adequately ended.
Author Response: The sentence has been revised and completed.
At line 230: “X” should be “X.”.
Author Response: Punctuation has been adjusted.
At lines 273-274: I don’t agree that the models used for the baseline data have ‘moderate to high’ marginal and conditional values.
Author Response: This mistake has been addressed. This comment was in reference to the values for the health check rather than the baseline data.
At line 411: “Afr. J. Herpetol.” should be italicized.
Author Response: Italicisation has been adjusted.
At lines 460-463: the two references are not cited in the text?
Author Response: The references have now been cited in the discussion to address the editor’s concern about interaction between frogs within a tank. The references have been adjusted accordingly.
Round 2
Reviewer 3 Report
No further comments.
Author Response
Decision: Accept after minor revision
Reviewer comment: Dear authors, Thank you for your recent edits. One issue still remains largely unresolved and that is the comment that pertains to the observer. Previously we asked for more detail on the observers collecting the behavioural data. How many people were used, what was their level of experience, how were they trained, what measures of reliability were used (e.g. inter-observer reliability) to assess the quality of the data? The manuscript has now noted that a single observer was used, but that does not allow the data quality to be assessed. Even with a single observer, they should be adequately experienced with the species, or be adequately trained in behavioural assessment, and their codings should be checked against another person who is trained, experienced etc. to ensure that their interpretation of behaviour is replicable. So the question still remains that if a single observer was used, and their training was restricted to training in the software (which is outlined in the manuscript), how can we know the quality of the dataset, and whether the results accurately reflect the behaviour of the animals that was shown during the study? Given the behavioural focus of the study, this section requires more detail and attention in the methods section. We look forward to seeing the revised manuscript.
Author Response: The authors thank the editor for providing more detail regarding this concern. As a result of the questions and comments from the editor, more detail about the observer’s experience, training and qualifications.
“At the time of study, the observer was an MRes Biological Sciences student (graduated October 2022 with Merit); the observer has experience of behavioral study in a range of taxon, gained through a BSc Animal Behavior and Welfare and experience in the zoological industry. The observer has experience recording the behavior of other Xenopus species and was trained in doing so by members of the Amphibian Behaviour and Endocrinology Group at the University of Chester. The observer received additional training relevant to X. longipes on section prior to the study with staff who work with the species professionally.”